# Behavioural plasticity and the transition to order in jackdaw flocks

Hangjian Ling [1,2], Guillam E. McIvor [3], Joseph Westley[3], Kasper van der Vaart[1], Richard T. Vaughan[4], Alex Thornton [3]* & Nicholas T. Ouellette [1]*

Collective behaviour is typically thought to arise from individuals following fixed interaction rules. The possibility that interaction rules may change under different circumstances has thus only rarely been investigated. Here we show that local interactions in flocks of wild jackdaws (*Corvus monedula*) vary drastically in different contexts, leading to distinct group-level properties. Jackdaws interact with a fixed number of neighbours (topological interactions) when traveling to roosts, but coordinate with neighbours based on spatial distance (metric interactions) during collective anti-predator mobbing events. Consequently, mobbing flocks exhibit a dramatic transition from disordered aggregations to ordered motion as group density increases, unlike transit flocks where order is independent of density. The relationship between group density and group order during this transition agrees well with a generic self-propelled particle model. Our results demonstrate plasticity in local interaction rules and have implications for both natural and artificial collective systems.

[1] Department of Civil and Environmental Engineering, Stanford University, Stanford, CA 94305, USA. [2] Department of Mechanical Engineering, University of Massachusetts Dartmouth, North Dartmouth, MA 02747, USA. [3] Center for Ecology and Conservation, University of Exeter, Penryn, Cornwall TR10 9FE, UK. [4] School of Computing Science, Simon Fraser University, Burnaby, B.C. V5A 1S6, Canada. *email: alex.thornton@exeter.ac.uk; nto@stanford.edu

Organisms ranging from bacteria to insects, fish, birds, and mammals, including humans, often behave collectively, producing spectacular, cohesive global patterns[1,2]. Physics-based self-propelled particle models[3–7] in which agents obey simple local interaction rules can generate similar group behaviour, including the emergence of ordered motion at high density[8–11]. It is thus often tacitly assumed that these rules are fixed for a given species. Recently, however, researchers have begun to observe that group properties such as size, density, and polarization can vary with external conditions[12–17], and that heterogeneity among group members can influence group dynamics[18–22]. These findings suggest that the interaction rules themselves may show plasticity. To date, empirical data to test this hypothesis has been very limited, as the vast majority of studies[23–33] have been performed in a single ecological context. Some support comes from laboratory experiments[13,15] showing that fish in small groups may modulate rule parameters, such as the size of their repulsion zone[13] or the tendency to initiate or follow movements[15] based on food availability and predation risk. Nevertheless, it remains unknown whether animals can switch between fundamentally different types of rules[1], such as between metric[3] and topological interactions[7] (interacting with all neighbours within a fixed distance or with a fixed number of individuals regardless of physical distance, respectively). Moreover, the potential for plasticity in interaction rules has seldom been tested for large groups of animals in the wild facing ecologically relevant challenges[34–36].

Bird flocks are one of the most extensively studied examples of collective animal behaviour. Our current understanding of local interactions in birds is based largely on flocks of species like starlings[28,30], pigeons[31,35–38], jackdaws[18,32], and chimney swifts[23]. Starlings in murmurations[28] and jackdaws in transit flights[18] have been found to interact topologically. In contrast, chimney swifts circling and landing on chimney roost sites[23] were reported to follow metric interactions. Different types of interaction rules are expected to have significant implications for the structure and function of flocks. For instance, numerical models[3,7] have argued that groups of birds interacting metrically would be expected to transition from disordered motion at low density to ordered flocking at higher densities. Such a density-dependent ordering transition has been experimentally confirmed in marching locusts[8] and migrating cells[9], but not in flocking birds. In contrast, models[7] find that topologically interacting groups should always display order regardless of density, thus enhancing group cohesion and response to external perturbations such as predators[28]. It remains unknown, however, whether interaction type is species-specific or whether a single species may switch between metric and topological interactions in different ecological contexts to optimize group function. Here, by measuring the three-dimensional (3D) movements of flocking jackdaws (*Corvus monedula*), we demonstrate that the local interaction rules and group-level properties can indeed change in different contexts.

## Results

**Local interactions in jackdaw flocks**. Here, using a high-speed 3D imaging system[39] (see Methods), we compare local interactions in wild flocks of jackdaws in two different ecological contexts. During the winter, jackdaws form large transit flocks as they return to their roosting sites in the evening. We recorded 6 transit flocks containing between 25 and 330 individuals (Supplementary Table 1; Supplementary Movie 1); an example is shown in Fig. 1a. Like many other species, jackdaws also commonly come together to inspect and drive away predators, typically in response to anti-predator recruitment calls known as "scolding" calls[40]. We

mimicked such events experimentally by using playbacks of scolding calls to induce groups to gather around a terrestrial model predator (see Methods for more information). We recorded 10 such "mobbing flocks" containing between 4 and 120 individuals (Supplementary Table 1; Supplementary Movies 2 to 3); an example is shown in Fig. 1b.

We find that the local interactions in the two types of flocks vary drastically: jackdaws use topological interactions in transit flocks, but metric interactions in mobbing flocks. To demonstrate this difference, we first consider the alignment angle $\theta$ between a focal bird and its neighbours as a function of topological rank $n$ and physical distance $r$, $\theta = g(n, r)$. For flocks with topological interactions, $\theta = g(n, r) = g(n)$, while for flocks with metric interactions, $\theta = g(n, r) = g(r)$. Thus, one way to test whether flocks use topological or metric interactions is by fixing $n$ and calculating the dependence of $\theta$ on $r$ alone, and vice versa.

In mobbing flocks, we find that $\theta$ increases considerably with $r$, and profiles with different $n$ nearly overlap (Fig. 1c). Thus, $\theta$ is primarily a function of $r$ alone. One may imagine that birds in mobbing flocks might fly in circles centred above the model predator, and that the increase of alignment angle with $r$ is simply due to geometric effects or external influences. We find, however, that this is not the case: birds in mobbing flocks are most likely to be found directly above the model predator, arguing against circling flight (Supplementary Fig. 1c). Moreover, the velocity component in the direction towards or away from the predator is of the same order of magnitude as the full velocity (Supplementary Fig. 1d), and the magnitude of the alignment angle is independent of the distance between the focal bird and the model predator (Supplementary Fig. 1e). Therefore, the alignment at small $r$ is due to local interactions between birds, and the increase of $\theta$ with $r$ indicates that these local interactions weaken as the separation distance increases.

In contrast to mobbing flocks, neighbours in transit flocks remain well aligned regardless of $r$ and $n$, since $\theta$ has little dependence on either $n$ or $r$ (Fig. 1c). Although the curves also have weakly positive slopes, the magnitudes of $\theta$ are much smaller than those in mobbing flocks. Since the transit flocks as a whole move towards the roost sites with high polarization (Supplementary Table 1), all the birds in the flocks share a common external influence. Nevertheless, transit flocks do exhibit large velocity fluctuations that are spatially correlated (Supplementary Fig. 2), indicating that they are distinct from the simple case of many birds flying in straight lines and instead have complex dynamics arising from local interactions between neighbours[4]. Thus, the strong alignment regardless of $r$ and $n$ is likely due to the combined effect of local interactions and external factors. It is difficult to separate the effects of local interactions and external factors on alignment angles. Thus, to more clearly discriminate between the interaction rules in mobbing and transit flocks, we consider the spatial distribution of neighbours relative to a given focal bird.

Analyses of the anisotropy of these distributions (that is, the extent to which the position of a neighbour relative to a focal bird is non-random (see Methods)) provide a critical test of whether interaction rules differ across contexts. Given that both mobbing and transit flocks show similar side-by-side neighbour distributions despite differing external factors, local anisotropy is expected to result from local interactions rather than external influences. We characterised the anisotropy by computing an anisotropy factor $\gamma = f(n, r)$ of the distribution of neighbours relative to a focal bird (see Methods). Here, a larger value of $\gamma$ indicates that the locations of the neighbouring birds relative to the focal bird are less random (i.e., more anisotropic). Previous studies[18,28] have shown that $\gamma$ reduces with increasing distance from the focal bird, and have defined the interaction range as the

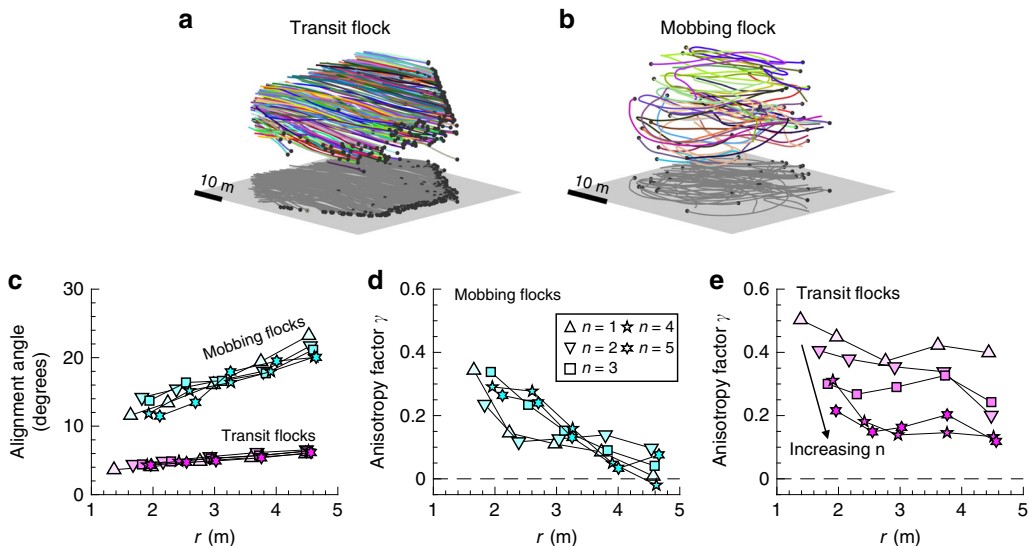

**Fig. 1** Metric interactions in transit flocks and topological interactions in mobbing flocks. **a** Jackdaw trajectories in a sample transit flock. **b** Jackdaw trajectories in a sample mobbing flock. **c** Alignment angle between a focal bird and its neighbours. **d, e** Anisotropy factor $\gamma$ for the distributions of neighbours relative to a focal bird for **d** mobbing and **e** transit flocks. $r$ is the distance between the focal bird and its neighbour, and $n$ is the topological rank. In **a, b** coloured lines show the jackdaws' three-dimensional (3D) trajectories, and the black dots show their positions in flocks at the final time step. The grey planes are arbitrary horizontal planes (that is, perpendicular to the direction of gravity). Data in **c, d** are calculated from 10 mobbing flocks and 6 transit flocks. Standard errors are smaller than the symbol size

distance where $\gamma$ reduces to its isotropic value, indicating that neighbours are distributed at random. We find that for mobbing flocks, $\gamma$ decreases significantly for larger values of $r$, while the profiles for different $n$ nearly collapse (Fig. 1d), indicating that $\gamma$ is primarily a function of $r$. In contrast, for transit flocks, $\gamma$ changes more rapidly with $n$ than with $r$, indicating that $\gamma$ depends primarily on $n$ rather than on $r$. These results provide strong evidence that jackdaws in mobbing flocks use metric interactions but switch to topological interaction in transit flocks. Jackdaws in mobbing flocks have a metric interaction range of about $r = 5$ m (roughly 14 body sizes), and in transit flocks typically interact with 7 to 8 neighbours, as shown in our previous study[18]. Thus, our observations provide strong empirical support for the plasticity of local interactions under different ecological contexts.

A final way to confirm that the interactions in transit flocks are topological is to show consistent behaviour of $\gamma = f(n)$ for flocks with different group densities, similar to the results shown in Ballerini et al. [28]. We thus measured $\gamma = f(n)$ for transit flocks #01, #03 and #04 where the group sizes are the largest among the six transit flocks. Although the three flocks have different densities (Supplementary Fig. 3b), profiles of $\gamma = f(n)$ for $n < 5$ nearly overlap (Supplementary Fig. 3a), confirming topological interactions. The $\gamma$ profiles for $n > 5$ do not overlap as well, likely due to edge effects as the sizes of these transit flocks are relatively small.

**Relationship between group density and group order**. The two types of local interactions produce distinct group behaviour: the metric interactions in mobbing flocks imply that order within groups depends on the group density, while the topological interactions in transit flocks generate highly ordered motion regardless of density. To characterize the emergence of order in mobbing flocks, we analysed the order parameter $\phi$ and the density $\rho$ for a total of 154 sub-groups (Supplementary Data 1) taken from the 10 mobbing flocking events recorded. Since birds frequently entered and left the measurement volume while recording during mobbing events, each recording contains

multiple distinguished groups separated in time (group selection criteria are given in the Methods). For each group at a given time $t$, we calculated the instantaneous order parameter $\phi_t = \langle v_i/|v_i| \rangle$ and density $\rho_t = 6 N/(\pi \langle d_i \rangle^3)$, where $v_i$ is the velocity of bird $i$, $N$ is the group size (i.e., the number of birds in the group), $d_i$ is the metric distance from bird $i$ to its most distant neighbour, and $\langle \rangle$ denotes an average over all birds. $\phi$ and $\rho$ are obtained by averaging $\phi_t$ and $\rho_t$, respectively, over all frames. At low density, the groups resemble disordered swarms (Fig. 2a) and $\phi_t$ fluctuates significantly (Fig. 2d). At moderate density, the flocks exhibit some degree of coherence (Fig. 2b); but at high density, all the jackdaws move and turn as a single cohesive unit (Fig. 2c) and $\phi_t$ remains close to 1 (Fig. 2d). Moreover, we find that $\phi$ increases monotonically with $\rho$ following the power law $\phi \sim \rho^{0.37}$ (Fig. 3), consistent with the classical self-propelled particle model of Vicsek et al.[3]. This agreement indicates that mobbing flocks are self-organized and that $\phi$ is an emergent property stemming from the local interactions—but that, in this case, those interactions are metric. In contrast, for transit flocks, sub-groups are nearly perfectly polarized regardless of the density (Fig. 3; Supplementary Fig. 4), likely due to the combined effect of topological interactions and environmental influences. Functionally speaking, topological interactions in transit flocks are very effective for maintaining group cohesion and enhancing awareness of and responsiveness to potential predators during long-distance travel[28]. Metric interactions in mobbing flocks, in contrast, may enable individuals to behave more independently, allowing them to focus more attention on the predator than on tracking distant flockmates.

**Self-propelled particle model**. Estimating the critical density for the transition to ordered motion is a key parameter for predicting the collective properties of the group[8–10]. We find that the critical density in mobbing flocks is close to 0 (Fig. 3), and show that this result can be mimicked by a self-propelled particle model (a modified Vicsek model[3]; see Methods) in the limit of vanishing noise $\eta$, a parameter that accounts for imperfect sensing. We ran the model for $N$ ranging from 5 to 200 while keeping a fixed

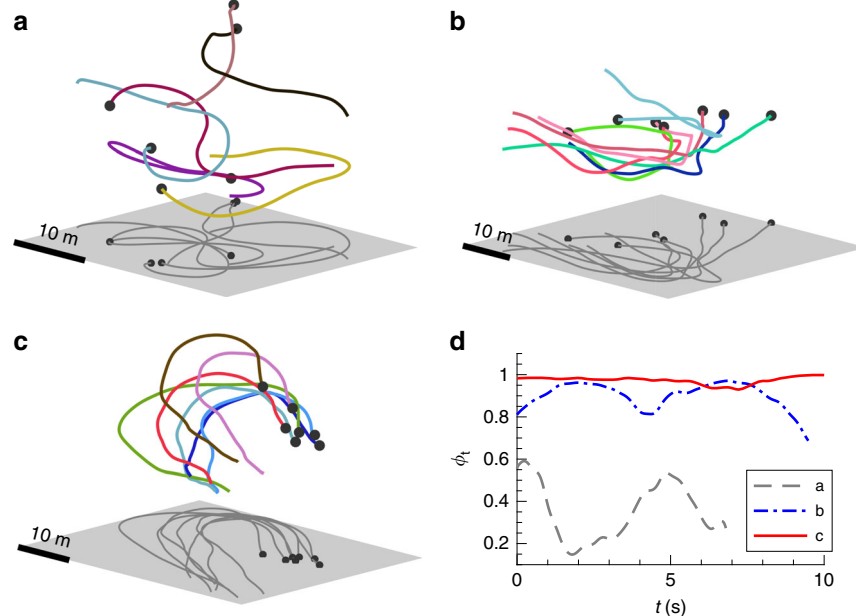

**Fig. 2** Three representative mobbing flocks showing increasing group order with group density. **a** Flock at low density ($\rho = 0.002\ \mathrm{m}^{-3}$). **b** Flock at moderate density ($\rho = 0.008\ \mathrm{m}^{-3}$). **c** Flock at high density ($\rho = 0.014\ \mathrm{m}^{-3}$). **d** Instantaneous group order $\phi_t$ for the flocks shown in (**a–c**). For **a**, **c** the meanings of the coloured lines, black dots, and grey planes are same as those shown Fig. 1

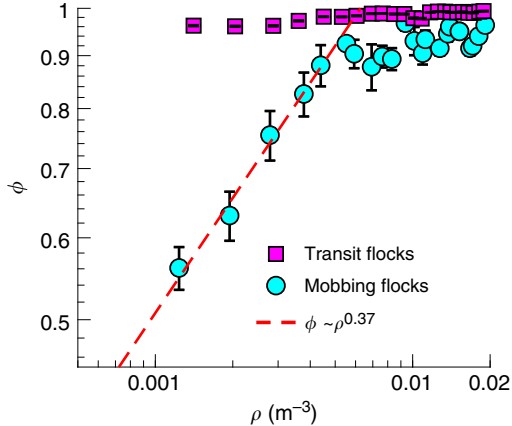

**Fig. 3** Relationship between group density and group order. For the mobbing flocks, results are calculated by averaging the results from 154 groups of jackdaws over a total of 36,960 time frames. Error bars represent standard errors. Distributions of group density $\rho$ and group order $\phi$ for the 154 groups can be found in Supplementary Fig. 6. For the transit flocks, results are calculated by averaging 30,498 samples, where each sample represents a local subgroup of one focal bird and four nearest neighbours embedded in a larger flock. Data for subgroups with sizes of 10 and 20 for transit flocks are shown in Supplementary Fig. 6

simulation volume. Thus, $N$ is proportional to the group density. The effects of $N$ and $\eta$ on the particle trajectories and group order are shown in Fig. 4. For small $N$, $\phi_t$ fluctuates significantly and the particle trajectories are qualitatively similar to our mobbing flocks at low density (Fig. 4a–c). For large $N$, $\phi$ is very close to 1 and the particle trajectories are similar to mobbing flocks at high density (Fig. 4b, c). $\phi$ increases smoothly with increasing $N$ (Fig. 4d), and both $\phi$ and $N-N_c$ obey power-law relations similar to our observations (Fig. 4e). Here, $N_c$ denotes the critical group size for the ordering transition (determined by the best-fit power law). In the limit of vanishing $\eta$, no disordered state exists (Fig. 4d) and $N_c$ approaches zero (Fig. 4f). Therefore, the nearly zero critical

density we observed for our mobbing flocks can be explained by a model where the alignment error is very low.

To test whether the ordering transition observed in our simulation is driven by metric-based local interactions and not by the introduction of a model predator, we ran the same simulations but with topological interactions rather than metric interactions. Results for a topological interaction range of eight neighbours show that the flocks remain highly order regardless of group size and density (Supplementary Fig. 5). Thus, we confirm that the ordering transition in the metric-based model is not due to the interaction with the predator.

## Discussion

Our results demonstrate that local interactions—and consequently group dynamics and morphology—vary considerably even within the same species in different ecological contexts. This behavioural plasticity may have major adaptive significance: rather than being limited to fixed interaction rules, animals may adjust their response to neighbours in different contexts to robustly optimize their group function and maximise fitness benefits[34]. Thus, it is crucial for future studies to consider ecological context when modelling and predicting animal movements in new and variable environments, as well as to understand the sensory and cognitive mechanisms underpinning behavioural plasticity. Our results may also find application in designing autonomous robotic swarms to respond to environmental cues by changing their interaction rules to perform different tasks[41,42]. Nevertheless, fully understanding and predicting the effect of ecological context on group movement and function will require additional study to separate the distinct effects of local interactions and external factors.

## Methods

**Study system.** Jackdaws (*Corvus monedula*) are a highly social, colony-breeding corvid found throughout much of the Western Palaearctic. At our study sites in Cornwall, more than 2000 jackdaws are fitted with unique colour ring combinations for individual identification. During the winter months, in the early evenings jackdaws form large polarized 'transit' flocks while travelling from their foraging grounds towards their roosts (often with staging stops at pre-roost trees) where they spend the night[43]. Jackdaws also form 'mobbing' flocks in response to

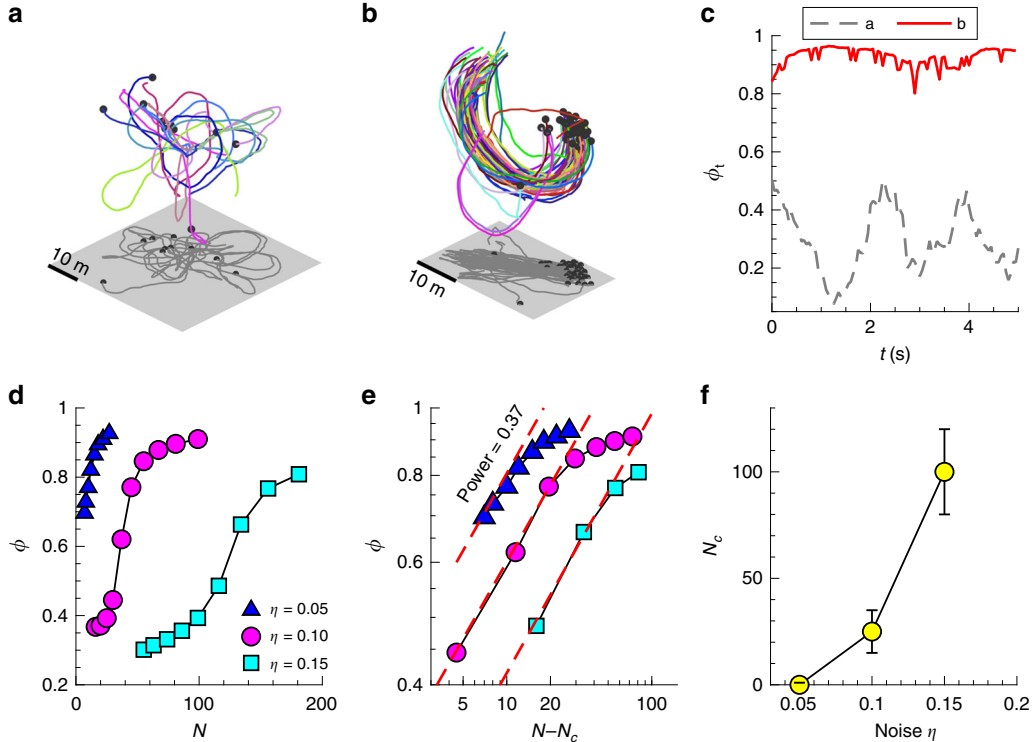

**Fig. 4** Self-propelled particle model captures the phase transition in mobbing jackdaw flocks. **a** A sample modelling result at low density. **b** A sample modelling result at high density. **c** Time variation of the instantaneous group order $\phi_t$ for the cases shown in (**a**, **b**). **d** Group order $\phi$ as a function of group size $N$ at three different noise levels. **e** $\phi$ as a function of $N-N_c$. **f** $N_c$ as a function of noise. Here, $N_c$ denotes the critical group size for the ordering transition. For **a**, **b** the meanings of the coloured lines, black dots, and grey planes are the same as those shown Fig. 1

distinctive scolding calls[40]; coming together to inspect and drive away aerial and terrestrial predators, such as raptors and foxes. Comparing the transit and mobbing flocks allows us to study whether local interactions are fixed or flexible in different ecological contexts.

**Three-dimensional imaging system**. To measure the movements of birds in 3D space, we used a high-speed 3D imaging system developed in our previous study[39]. The system included four hardware synchronized, high-speed USB-3 cameras (Basler ace acA2040-90 um, pixel size of 5.5 μm, sensor resolution of 2048 by 2048 pixels) with overlapping fields of view. The cameras were seated on the ground, pointed toward the sky, and were separated by a maximal distance of about 50 m. The imaging lenses (Tamron, M111FM08) had a focal length of 8 mm and an angle of view of 71°. At a height of 50 m, the imaging area was 60 by 60 m² and the imaging uncertainty was 0.04 m, which allowed the precise detection of bird positions. Two laptops (ThinkPad P51 Mobile Workstation) were used for data storage and allowed continuous imaging at 60 frames per second for up to 200 s.

The camera calibration followed a similar procedure to that developed by Theriault et al. [44]. We imaged two balls of distinguishably different sizes (10 and 12 cm) carried by a drone that was flown through the measurement volume. The 2D positions of the balls identified on each camera provided calibration points. About 300 calibration points were used to determine the camera parameters. The distance between the two balls (fixed at 1.0 m) provided a physical scale for the calibration. The calibration error (defined as the root-mean-square distance between the originally detected 2D coordinates and those generated by re-projecting 3D points onto the 2D image planes of the cameras) was less than 0.5 pixels.

After recording the image data, we reconstructed the trajectories of individual birds in 3D space. First, we calculated birds' 2D locations on the images based on intensity-weighted centroids (noting that these 2D locations may not represent body centres due to the flapping wings). We matched the 2D coordinates across different views by searching within a small tolerance around the projected epipolar lines from other cameras. 3D locations were then calculated using a least-squares solution of the line-of-sight equations[45]. In addition, we solved the optical occlusion problem by associating every detected 2D coordinate with a 3D position[39]. Finally, 3D locations of the same object in multiple time frames were linked together based on a three-frame predictive particle tracking algorithm[46]. We applied a Gaussian smoothing and differentiating kernel[47] to the 3D trajectories to obtain accurate velocities and accelerations.

Since we initially calculated the birds' 2D positions based on the intensity-weight centroids, the resulting 3D trajectories included a high frequency

component caused by wing motions. To remove this high frequency signal, we applied a low-pass filter to the measured acceleration and then integrated the filtered acceleration to obtain the motions corresponding to bird's centre of mass. Along each bird's trajectory, we measured the position, velocity, and acceleration in a Cartesian coordinate system. In our previous study[39], the wing motion and wingbeat frequency were also calculated. Here, we only report statistics related to a bird's centre of mass.

**Data collection and event selection criteria**. Jackdaws often fly together with rooks (*Corvus frugilegus*)[43], forming mixed-species flocks. To avoid any effects caused by species differences, we only selected flocking events in which all the birds were jackdaws (identified by vocalisations and morphological characteristics) for both transit and mobbing flocks. We also required flock images to be captured by all four cameras.

First, we recorded 6 transit flocks over the period from December 2017 to March 2018. To capture the transit flocks, we set up the imaging system along the typical flight paths of flocks such that the birds flew directly over the camera array. The imaging locations were in the vicinity of winter roosts near Mabe and Gwennap, Cornwall, UK. We required the flocks to be moving primarily in one direction without making large-scale turns and the time durations to be longer than the time scale for birds to exchange neighbours, so that our tracking results represented typical flock movement. This neighbour exchange time scale was less than 2 s[18]. Statistics for the 6 flocks including time duration, group size, nearest neighbour distance, and flight speed are given in Supplementary Table 1. Videos of the flocks are provided in Supplementary Movie 1.

We recorded 10 mobbing flocks at our nest-box colonies near Stithians, Cornwall, UK, during the 2018 breeding season (May to July) when the jackdaws in our study colonies remained in the vicinity of their nest-boxes (these individuals are known to join winter flocks flying to the Gwennap roost). To induce collective anti-predator responses, we used a taxidermy fox (*Vulpes vulpes*) holding a remote-controlled, flapping bird resembling a jackdaw in its mouth. The fox was initially hidden under a sheet in an open field, in the centre of the camera array. Once the fox was uncovered and the experimenter had returned to their hide, we broadcast pre-recorded scolding calls through a remote-controlled FoxPro loudspeaker placed in a hidden position on the ground beside the fox [for details of recording protocols, see Woods et al. [40]]. Playback tracks consisted of three bouts of 8 calls, each separated by 10 s. This mimicks naturally occurring calling bouts, and the amplitude was normalised across all tracks. As the magnitude of collective responses is influenced by the characteristics of the caller[40], we used only calls produced by colony members, which would be familiar to the birds in the vicinity.

Statistics for the 10 mobbing flocks including time duration, group size, nearest neighbour distance, and flight speed are also given in Supplementary Table 1. Videos of the flocks are provided in Supplementary Movies 2 and 3. Since birds frequently left and entered the measurement volume in mobbing flocks, we manually selected distinguishable groups of birds from each event that satisfied the criteria that (i) the group size was larger than 3; (ii) the time duration was longer than 1.5 s; (iii) the jackdaws did not leave the measurement volume during the selected time period; and (iv) there were no transitions from disordered to ordered states or from ordered to disordered states caused by, e.g. group fusion or fission. Based on these criteria, we selected 154 groups of birds (Supplementary Data 1). The statistics including time duration, group size, group density, and group order for the 154 groups are shown in Supplementary Fig. 6.

In a previous study[18], we showed that transit flocks recorded in the winter contain pairwise substructure that arises from the life-long monogamous pair bonds in jackdaw societies. Such pairwise structures are also present in all 6 transit flocks in the current study. For the transit flocks, we calculated the joint probability density functions (PDFs) of the distance to nearest neighbour ($D^{n=1}$) and the distance to the second nearest neighbour ($D^{n=2}$). These joint PDFs show two distinct regions of high probability (Supplementary Fig. 7), with one region where $D^{n=1}$ is nearly constant regardless of $D^{n=2}$ indicating the presence of paired birds. However, for the mobbing flocks recorded during the summer, the joint PDFs of $D^{n=1}$ and $D^{n=2}$ only show one region of high probability (Supplementary Fig. 8) where $D^{n=1}$ increases linearly with $D^{n=2}$. This indicates that the mobbing flocks contain no pairs. The likely explanation for this difference is that the mobbing experiments were conducted during the breeding season, when both members of the pair are foraging independently so as to maximise the rate at which their young are provisioned.

**Ethical note**. All field protocols were approved by the Biosciences Ethics Panel of the University of Exeter (ref 2017/2080) and adhered to the Association for the Study of Animal Behaviour Guidelines for the Treatment of Animals in Behavioural Research and Teaching.

**Anisotropy factor of neighbour structure**. Following a method developed in our previous study[18], we calculated the anisotropy factor $\gamma$ of the first nearest neighbour distribution relative to a focal bird. We first determined the position of a neighbouring bird relative to a focal bird as $d\mathbf{x} = \mathbf{x}^{\text{neighbour}} - \mathbf{x}^{\text{focal}}$, where the superscripts 'focal' and 'neighbour' denote quantities for the focal and neighbour birds, respectively. We then translated the two components of $d\mathbf{x}$ located in the horizontal plane to a new coordinate system $(\xi, \zeta)$ where $+\xi$ was aligned with the flight direction of the focal bird. Here, we ignored the velocity component in the gravity direction, since it is typically much smaller than the horizontal components (Supplementary Fig. 9). Then, we normalized the vector $(\xi, \zeta)$ to create a unit vector, denoted as $(d\xi, d\zeta)$. The anisotropy factor was defined as $\gamma = <d\zeta d\zeta - d\xi d\xi>$, where $<>$ denotes an average over all birds within the group. The value of $\gamma$ ranges from −1 to 1 by construction. $\gamma > 0$ indicates that the neighbouring bird is more likely to be next to the focal bird, $\gamma < 0$ indicates that the neighbouring bird is more likely to be in front or back, and $\gamma = 0$ indicates an isotropic structure.

**Self-propelled particle model**. To mimic the phase transition observed in the jackdaw flocks, we used a self-propelled particle model where particles followed simple local interaction rules based on that of Vicsek et al. [3]. In this model, all particles move at a constant speed $U_0$ and align their directions of motion to the average velocity of all neighbours located within a metric distance, $r_0$, with some noise added. The Vicsek model was originally developed in two dimensions, and later extended to 3D[48]. Here, we used the 3D version of this model. We also added a repulsion zone[49] with radius $r_{\text{rep}}$ to prevent particles from forming locally dense clusters. To simulate the effect of the external stimulus that kept the mobbing flocks within a certain area, we introduced an additional potential well (here, a harmonic force) felt by every particle. This procedure followed that used by Attanasi et al. [24] who simulated swarms of midges that were stationary with respect to a ground marker. With such a potential well, particles are guaranteed to be confined within a certain volume without the need for periodic boundary conditions or long-range attraction forces.

Specifically, the velocity of each particle $i$ at the next time step was given by

$$\mathbf{v}_i(t + \Delta t) = U_0 \cdot \mathbf{\Omega}_\eta \{\Theta[\sum_{j \in S_i} \mathbf{v}_j(t) - \beta \mathbf{x}_i(t)]\} \tag{1}$$

where $S_i$ is the spherical volume centred on particle $i$ with a radius of $r_0$, the operator $\Theta$ normalizes the argument to be a unit vector, and the operator $\mathbf{\Omega}_\eta$ rotates the argument vector by a random angle chosen from a uniform distribution with maximum amplitude of $4\pi\eta$. Note that the term $-\beta \mathbf{x}_i$ is the harmonic force that pushes particle $i$ back towards the origin. The strength of this harmonic force is controlled by $\beta$. The position of particle $i$ at the next time step is then given by $\mathbf{x}_i(t + \Delta t) = \mathbf{x}_i(t) + \mathbf{v}_i(t + \Delta t) \cdot \Delta t$.

We ran this model in a 3D sphere with a radius of $R$ and open boundary conditions. Initially, particles were randomly distributed within the sphere and moved in random directions. The parameters used in our models were $R = 50$ m, $U_0 = 10$ m/s, $r_0 = 4$ m, $\Delta t = 0.05$ s, $r_{\text{rep}} = 0.5$ m, and $\beta = 0.04$. The values of these parameters are selected according to the observation of real bird flocks. For

example, birds in mobbing flocks have flight speed close to 10 m/s (Supplementary Table 1), a repulsion zone of their body size, and an interaction range of about 5 m (Fig. 1d). Increasing $\beta$ causes birds to flying closer to the predator. We selected $\beta = 0.04$ such that the trajectories of birds cover the simulated domain size. We varied the number of particles $N$ from 5 to 200 and the noise level $\eta$ from 0.05 to 0.15 to study their effects on the group order. At each given $N$ and $\eta$, we ran the model 50 times, and selected 100 time frames between $10^4 \Delta t$ to $10^6 \Delta t$ with an interval of $10^2 \Delta t$ for analysis. Thus, the group order at a given $N$ and $\eta$ was calculated by averaging 5000 samples.

## Data availability
Raw images captured by one of the four cameras and the reconstructed 3D movement trajectories of the jackdaws are provided in Supplementary Movies 1 to 3. The group density and group order for 154 groups are provided in Supplementary Data 1. Plain text files, each including bird ID number, position, time, and velocity at every time step are provided in Supplementary Data 2 to 4. All data required to reproduce the results in this study are included in Supplementary Data 1 to 4. The data analysis codes (including the self-propelled particle model) to generate all figures in the paper are also provided. Supplementary Data, Supplementary Movies, and data analysis codes are available at: https://figshare.com/s/472d354cc9e823a8f48f

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

## Acknowledgements

This work was supported by a Human Frontier Science Program grant to A.T., N.T.O. and R.T.V., Award Number RG0049/2017. We are grateful to Paul Dunstan, Richard Stone, and the Gluyas family for permission to work on their land, and to Victoria Lee, Beki Hooper, Amy Hall, Paige Petts, and Christoph Petersen for their assistance in the field.

## Author contributions

N.T.O, A.T., and R.T.V. conceived the ideas; G.E.M. and A.T. designed the mobbing experiment; G.E.M. and J.W. collected the data; H.L., N.T.O, and K.V. analysed the data; H.L. and N.T.O developed the model; all led the writing of the paper. All authors contributed critically to the drafts and gave final approval for publication.

## Competing interests

The authors declare no competing interests.
