## [Peer Review File · Nature Communications]

Reviewers' comments:

Reviewer #1 (Remarks to the Author):

This paper addresses the interesting and important question of whether the same species in different ecological contexts could show fundamentally different interaction rules, using jackdaw transit and mobbing flocks as the comparison. The authors measure several quantitative properties of the different types of flocks and use these to make the claim that mobbing flocks show metric interactions whereas transit flights show topological interactions. While the idea is interesting and important, and the data collection effort impressive, I found that the analytical methodology had some fundamental flaws that led me to be very skeptical of the paper's conclusions. I will elaborate on these points below.

1. Alignment vs. distance (Fig 1c). The data show that the angle between two birds' headings increases as a function of the distance between the birds in mobbing flocks, but remains relatively constant (and low) in transit flights. The authors use this finding to support the claim that in mobbing flocks the birds are interacting metrically, whereas the topological interactions allow for longer-range alignment in the transit flights (L70-75). However, there is a much simpler explanation that is completely ignored: in the mobbing flights the birds are generally flying in a circle (around a model predator) whereas in the transit flights they are flying straight (in the direction of the roost). Indeed, the authors have even excluded data from transit flights in which the flocks turned too much (L240). It is a basic geometric fact that if you have objects moving a circle, then their headings will be less aligned the farther apart they are, whereas if they are moving in one direction they will be aligned regardless of distance. In both cases, the result is driven by the overall geometry of the flock (which in this case is determined by external and not self-organized factors, i.e. the predator model and the roost direction). Therefore I don't find that this convincingly says anything about metric vs topological rules.

2. This relates to a second important point that is not addressed in the paper - external influences. In both contexts, the birds are interacting not only with one another, but also with some external factor i.e. the predator model or the roost. However the analyses do not take these influences into account, and treat all patterns as if they are emerging from self-organized processes. It is of course difficult to disentangle the effects of inter-bird interactions and interactions with external factors, however in this case I think the authors would have some chance of accomplishing this because they actually know what the external influences are (in contrast to some instances where these are unknown) - so this could actually be a great opportunity! I would therefore suggest much more exploration of these factors - for instance the analyses could also take into account the relative positions of the birds relative to the stimulus in the mobbing flights. In the model of mobbing flights, the authors do include a term for the influence of the predator model, however this is somewhat unsatisfactory because its influence is not really usefully disentangled from the local interactions (see point 5 below).

3. Anisotropy analysis. I found this analysis a bit confusing. In the text, it is stated that the anisotropy was computed "for nearest neighbors as a function of r " (L82), and this is also echoed in the Figure 1 caption. Is it really that only the nearest neighbor is used, or is it for all neighbors as a function of r ? I think the latter would make much more sense (indeed I can't understand why doing the former would be correct at all), but from the text I couldn't tell which was done. Assuming the analysis was done for all neighbors and not just the nearest neighbor, was some

kind of binning done for different ranges of values of r ? Also, I think this analysis could also be quite affected by the circular vs aligned flock types, similar to my first point.

4. Testing topological models. As the paper is about distinguishing metric vs topological interactions, could it not also be informative to conduct the same analyses done as a function of metric distance for the topological case? In other words, compute the same properties, but as a function of the topological distance instead of the metric distance. This might actually be a better way to address the question of metric vs topological interactions, because one would expect that if topological interactions are really at play, these functions should be consistent across different flocks when you plot them as a function of topological distance, but not when you plot them as a function of metric distance (and vice versa, if there are really metric interactions).

5. Simulation models. The authors show that their data are in agreement with a theoretical model based on metric interactions (modified Vicsek model). However, they don't test the alternative model using topological interactions, so this leads to the question what is actually driving the agreement. The model also includes a term for the influence of the predator (which is good!), but this could easily be driving the effects they see. I would suggest that they compare two alternative models, both of which have the same interaction term for the predator, but one of which uses topological and the other uses metric interactions. Agreement with one or the other model would then support one or the other type of interactions. On another note related to the model, I wondered how the parameters were chosen, and whether the chosen values would affect the exponent of the "power law" (which I'm not particularly convinced is a power law anyway). Some sensitivity analysis would be useful here.

Minor comments:

L105-108: This is a far too strong interpretation. The authors are arguing that the flocks are aligned due to topological interactions, however it's equally plausible that they are just aligned because they are all heading to the same roost (and because the authors have excluded data from when they were turning...)

Fig 1. I think showing standard deviations would be more informative than standard errors because the number of data points is huge and the data points are not independent anyway, so dividing by N doesn't make much sense in this case.

Reviewer #2 (Remarks to the Author):

In the manuscript titled 'Behavioural plasticity and the transition to order in jackdaw flocks', the authors measure high-resolution trajectory segments from flocks of wild jackdaws in two different contexts: travelling and mobbing. From the data they show that the local rules that govern the interactions could be well described by two different types of models – the topological for transiting groups and the metric for mobbing – which results in fundamentally different collective behaviour on the group level. In the last 15 years or so, animal systems have been categorized into two different, mutually exclusive classes of models based on metric or topological interactions. By showing the plasticity that a single animal system can behave – based on context – like both of these two classes, this manuscript makes a relevant step into the direction to solve the seemingly conflicting situation. The experimental setup and the data collected are neat and well support the claims of the paper. I also applaud that the data are available in an easily accessible format. The manuscript is generally well written and clear to understand. I find the manuscript interesting, and I think it would worth the interests of readers outside of the specific field of collective behaviour too. I have only minor concerns, suggestions and comments.

There authors makes the claim at multiple places that the line of research that they do is novel, and has never been studied before (lines 13-14 in the abstract; or a variation of this in lines 39-

40, etc.). Although I accept that this is a relatively new direction, the statement in its current form is not true, given that there exists some research that fits this topic. See, for example, Zhang et al. 2014, where they show that pigeon flocks may switch hierarchical and egalitarian strategies during navigation, or Nagy et al. 2013, where the context dependence of the underlying interaction network is shown. This latter is also a counter-example to the statement "each of which has been studied in only one ecological context" at lines 42-45.

I find the sentence at lines 54-56 not clear enough, neither justified, and there is also some mismatch with the citations given there. Please rephrase.

Lines 96-100: I find the definition how the density was calculated puzzling. This may be a good approximation for small three-dimensional flocks – where most individuals are on the periphery of the group –, since the distance of the individual furthest from the focal individual defines the diameter of the inclusion sphere. But in case of larger flocks – where most individuals are inside the convex hull of the flock – one may think that the distance of the furthest individual rather defines the radius and not the diameter of the sphere. Moreover, if the shape of the flock may well be described as being flat, the density approximation (especially for larger flocks) may give misleading results. Could the authors comment on this? Could a local density calculation method be used instead to get values closer to the possible real density of the flocks?

Lines 107-109: I am not convinced by this statement. What is the cause here? The existence of a global direction or a goal to reach could result in the apparent local rules, and not the other way around, couldn't it?

Fig. 3 and the related supplementary figures: Why were there different ways of calculation used for the mobbing and transit groups? While the current method used gives the "global" order parameter for mobbing (although groups are smaller here), for the transit groups, the order parameter is calculated locally. I find that this difference makes it hard to compare the results of the two different contexts. Calculating the values of the local order parameter also for the mobbing the same way as it was performed for the transit group would be easy and would resolve this problem.

Lines 289-291: 2D may be a good representation for transit flocks but how about mobbing? Can the authors justify this for that case as well?

Minor corrections:

- The reference numbering seems a bit messy at line 33.
- References 8, 11, 39 seem corrupted.
- Fig. S3: How can there be values at $x < y$, meaning that the distance from the second nearest neighbour is less than the distance from the closest neighbour? Also, the scales of x and y axes are a bit confusing. Please use the same unit length and start both axes' range from 0.

As a consequence of all the above, I recommend minor revision and consider the manuscript a good candidate for being published in Nature Communications.

Response to Reviewer #1

The authors have now revised the paper in line with my suggestions and those of the other reviewer. I appreciate the efforts the authors have made to address the comments, and I find the paper much improved and more convincing. Overall, I feel that the authors have done a good job addressing the comments. The justification and description of the anisotropy analysis is now much clearer, the additional simulations are quite convincing, and I appreciate that the authors have now at least tried to address the potential circular flight bias. The only major criticism I still have is regarding the alignment angle analysis, described below.

Response: We thank the reviewer for the positive feedback. In this revision, we have focused on this final issue.

The additional analysis regarding alignment angle and attempting to control for possible circular flight is a nice addition to the paper. However, the authors' claim that this is evidence of topological interactions rather than a byproduct of circular trajectories really rests on Figure R1e, and I was still curious if this type of pattern could arise simply from circular flight. So I made a little simulation (R code copied below). In the simulation, I just distributed the birds around a predator at (0,0) and gave them roughly circular-directed velocities vectors, with some jitter in their positions, velocities, etc. The authors are welcome to play around with it. In general, I found it's not too difficult to get those curves to fall very close to one another, just from this circular flight model with some noise. Seeing this just made me a bit skeptical of the whole argument that one can tell what type of interactions one has based on this type of analysis.

Response: We appreciate the time the reviewer spent to build this simulation of circular flights and compare the outcomes to the empirical observations we report. However, we disagree that our argument that the increase of alignment angle with metric distance is not a byproduct of circular flight relies only on Figure R1e (now fig. S1e in the supplementary materials); rather, we draw our conclusions from the synthesis of several different pieces of evidence, such as those shown in Figures S1a-d. Critically, although the circular flight model proposed by the reviewer does indeed capture the trends in Figure S1e, it does not reproduce the results in Figure S1d—that is, that the magnitudes of the radial velocity of the birds relative to the model predator are on the same order of magnitude as the bird flight speed). In the model, the reviewer assumed that the bird velocity is $v_x \sim \sin(\theta + \text{noise})$ and $v_y \sim \cos(\theta + \text{noise})$. With those assumptions, the radial velocity v_r should be close to 0. However, as shown in Figure S1d, the radial velocity in real mobbing flocks is much larger than 0. Thus, a simple circular flight model can reproduce some of our results but not all of them—because birds in the real mobbing flocks are not simply flying in circles.

Our conclusion that mobbing flocks display metric interactions also relies on multiple pieces of evidence, not just the alignment angles. This evidence includes, for example, the anisotropy analysis (Line 102 to 119; Fig. 1d-e) and the relationship between group density and group polarization (Fig. 3).

To reflect these points, we added a new section in the supplementary materials:

“Supplementary discussion

To test whether our finding for mobbing flocks could potentially result simply from birds flying in circles around the model predator, we built a two-dimensional circular flight model and compared its results with the empirical observations (we thank an anonymous reviewer for this suggestion). In this model, the agent's velocity is prescribed as $v_x \sim \sin(\theta + \text{noise})$ and $v_y \sim \cos(\theta + \text{noise})$. Although this simple model can capture the results shown in Fig. S1e (that is, that the alignment angle increases with metric distance between agents), it does not reproduce the results in Fig. S1d (that the radial velocity of birds with respect to the predator is on the same order of magnitude as the flight speed).”

On a related note, the authors claim that the flatness of the curve for transit flocks with r (Fig 1c) indicates non-metric interactions. But I still think this feature could arise from the flock just being very polarized in general, and I don't see how the overall polarization necessarily indicates topological interaction rules. In addition, the curve is actually not completely flat and does seem to go up a bit, similar to the other curve but with a shallower slope. So by the authors' logic, couldn't this also be consistent with metric interaction rules?

Response: We do not claim that “the flatness of the curve for transit flock indicates non-metric interaction”; and indeed we agree with the reviewer that the strong alignment in transit flocks can be partially attributed to the overall polarization. Instead, our strongest evidence for topological interactions is the anisotropy analysis. That is why we state in the manuscript on Line 97 that: “...the strong alignment regardless of r is likely due to the combined effect of local interactions and external factors...”

In addition, we do realize that the curve for transit flocks is not flat (Fig. 1c). What we therefore emphasized in the paper is that the values are small (close to 5 degrees) even for large r . In contrast, in mobbing flocks, the values can be larger than 20 degrees for large r . In this case, we expect that the magnitude of the alignment angle is probably more important than the trends. To address the reviewer's concern, we have added the following sentence at Line 91: “...Although the curves also have weakly positive slopes, the magnitudes of the alignment angles are much smaller than those in mobbing flocks...”

I was trying to think about how to salvage this whole alignment angle analysis, and I wondered if one couldn't do something similar to in the anisotropy analysis, (which seems more compelling to me). In other words, what about computing the alignment angle as a function of metric distance and as a function of topological distance for flocks of different densities? The prediction (I think) would then be that for topological flocks the curves should collapse if you plot them vs topological distance and likewise for metric flocks and metric distance. I think one would still need to figure out a way to account for the fact that in the mobbing flocks the group has to turn back toward the predator (even if they aren't completely circular trajectories this is still a feature), and I don't really know how to do this. But could this perhaps be a way forward?

Response: It is important to note that our arguments for topological vs metric interactions rely primarily on analyses of anisotropy and not alignment angles. We calculated the anisotropy factor γ as a function of both topological rank n and metric distance r . We show that γ depends primarily on r for mobbing flocks, and depends primarily on n for transit flocks. Based on that, we conclude that interactions in mobbing flocks are metric and interactions in transit flocks are topological.

We also calculated the alignment angle as a function of n and r , as shown in Figure 1c. As expected, for mobbing flocks, the alignment angle increases significantly with r , and the profiles for different n overlap. This indicates that for mobbing flocks the alignment angle mainly depends on r , similar to the trends of γ . However, for transit flocks, the alignment angle remains small regardless of n and r . The reason, as correctly noted by the reviewer, is that the alignment angle in transit flocks can be attributed to both local interactions and external factors. Because it is difficult to distinguish between these two sources of alignment, to argue for topological interactions in transit flocks, we mainly rely on the anisotropy factor rather than the alignment angle.

To address the reviewer's concern, and make the importance of anisotropy analyses more explicit, we have modified the paragraphs in Lines 70 to 121 to read:

“We find that the local interactions in the two types of flocks vary drastically: jackdaws use topological interactions in transit flocks, but metric interactions in mobbing flocks. To demonstrate this difference,

we first consider the alignment angle θ between a focal bird and its neighbours as a function of topological rank n and physical distance r , $\theta=g(n, r)$. For flocks with topological interactions, $\theta=g(n, r)=g(n)$, while for flocks with metric interactions, $\theta=g(n, r)=g(r)$. Thus, one way to test whether flocks use topological or metric interactions is by fixing n and calculating the dependence of θ on r alone, and vice versa.

In mobbing flocks, we find that θ increases considerably with r , and profiles with different n nearly overlap (Fig. 1c). Thus, θ is primarily a function of r alone. One may imagine that birds in mobbing flocks might fly in circles centred above the model predator, and that the increase of alignment angle with r is simply due to geometric effects or external influences. We find, however, that this is not the case: birds in mobbing flocks are most likely to be found directly above the model predator, arguing against circling flight (Supplementary Fig. 1c). Moreover, the velocity component in the direction towards or away from the predator is of the same order of magnitude as the full velocity (Supplementary Fig. 1d), and the magnitude of the alignment angle is independent of the distance between the focal bird and the model predator (Supplementary Fig. 1e). Therefore, the alignment at small r is due to local interactions between birds, and the increase of θ with r indicates that these local interactions weaken as the separation distance increases.

In contrast to mobbing flocks, neighbours in transit flocks remain well aligned regardless of r and n , since θ has little dependence on either n or r (Fig. 1c). Although the curves also have weakly positive slopes, the magnitudes of θ are much smaller than those in mobbing flocks. Since the transit flocks as a whole move towards the roost sites with high polarization (Supplementary Table 1), all the birds in the flocks share a common external influence. Nevertheless, transit flocks do exhibit large velocity fluctuations that are spatially correlated (Supplementary Fig. 2), indicating that they are distinct from the simple case of many birds flying in straight lines and instead have complex dynamics arising from local interactions between neighbours⁴. Thus, the strong alignment regardless of r and n is likely due to the combined effect of local interactions and external factors. It is difficult to separate the effects of local interactions and external factors on alignment angles. Thus, to more clearly discriminate between the interaction rules in mobbing and transit flocks, we consider the spatial distribution of neighbours relative to a given focal bird.

Analyses of the anisotropy of these distributions (that is, the extent to which the position of a neighbour relative to a focal bird is non-random (see **Methods**)) provide a critical test of whether interaction rules differ across contexts. Given that both mobbing and transit flocks show similar side-by-side neighbour distributions despite differing external factors, local anisotropy is expected to result from local interactions rather than external influences. We characterised the anisotropy by computing an anisotropy factor $\gamma=f(n, r)$ of the distribution of neighbours relative to a focal bird (see **Methods**). Here, a larger value of γ indicates that the locations of the neighbouring birds relative to the focal bird are less random (i.e., more anisotropic). Previous studies^{18,28} have shown that γ reduces with increasing distance from the focal bird, and have defined the interaction range as the distance where γ reduces to its isotropic value, indicating that neighbours are distributed at random. We find that for mobbing flocks, γ decreases significantly for larger values of r , while the profiles for different n nearly collapse (Fig. 1d), indicating that γ is primarily a function of r . In contrast, for transit flocks, γ changes more rapidly with n than with r , indicating that γ depends primarily on n rather than on r . These results provide strong evidence that jackdaws in mobbing flocks use metric interactions but switch to topological interaction in transit flocks. Jackdaws in mobbing flocks have a metric interaction range of about $r=5$ m (roughly 14 body sizes), and in transit flocks typically interact with 7 to 8 neighbours, as shown in our previous study¹⁸. Thus, our observations provide strong empirical support for the plasticity of local interactions under different ecological contexts.

A final way to confirm that the interactions in transit flocks are topological is ...”

Response to Reviewer #2

I would like to thank the authors for their detailed responses and additional analyses. I believe this version is nicely improved and the authors have addressed my concerns. And I applaud the replies they wrote for the issues raised by the other reviewer too - with which I completely agreed. I suggest the manuscript to be accepted for publication.

Response: We thank the reviewer for the positive feedback.

Minor comment:

Supplementary Fig. 5: the local density (the actual density of the flock) looks similar in both examples given on panels a and b. I suggest changing the legend to small and large group size. Otherwise, please give an example where the local density of the flock is also low, similarly to the experimental data (and not only the density - calculated as the number of individuals divided by the total volume - is low).

Response: The legend was changed to read: “**Supplementary Fig. 5 | Model with topological interactions. a.** A sample modelling result at small group size ($N=10$). **b.** A sample modelling result at large group size ($N=50$). **c.** Time-variation of ϕ_t for the cases shown in **a** and **b**. **d.** Group order ϕ as a function of group size N at three different noise levels.”

Reviewers' Comments:

Reviewer #1:

Remarks to the Author:

Thanks to the authors for their responses and revisions regarding the point I raised in the previous review. I agree with them that the strongest evidence in favor of metric vs topological interactions in the two contexts comes from the anisotropy analysis, and as stated in my previous review I find this analysis well done and supporting of their claims. I do think they have slightly misinterpreted my previous point regarding the alignment angle analysis, which I tried to illustrate through a toy model simulation. The simulation I made was not meant to capture all of the features of the system, rather it was meant to show that the main piece of evidence the authors use to back their claims about topological and metric interactions with respect to alignment angle, i.e. Figure 1c, is possible to reproduce in a manner that doesn't involve a difference between metric and topological interactions but rather just results from a geometric effect of birds turning back toward the predator in one case and not the other. Of course I do not think that the birds are simply flying around in circles - this is obviously not true. My point is that flying around a focal point is different than flying straight when it comes to alignment angles, and that this turning effect could lead to similar results to those shown in Figure 1c. As a side note, the authors point out that my simulation doesn't capture the velocity distribution, and in particular the fact that the radial speed is of the same order of magnitude as the overall speed. Of course it doesn't, because this was a toy example meant to illustrate my point about Figure 1c, not to capture all aspects of the system. In fact I've set all bird speeds to 1 in the simulation, and depending on the noise added to the velocities, the radial speeds can range from close to 0 to close to 1.

Anyway, I think the authors have taken my main point that the anisotropy analysis is really the thing that can distinguish between metric and topological interactions in the paper, not the alignment angle analysis. I think we still disagree slightly on whether the alignment angle analysis is providing any insight into this issue (I think it really does not, the authors still seem to think it does). I'm fine with them leaving that analysis in, as long as they are clear about its limitations in the text. The sentences: "It is difficult to separate the effects of local interactions and external factors on alignment angles. Thus, to more clearly discriminate between the interaction rules in mobbing and transit flocks, we consider the spatial distribution of neighbours relative to a given focal bird." go in this direction, which I think is a good addition.

More generally, think that the question of how we can rule out effects of external influences on the inferences we draw regarding interaction type is an important one for this field - though a thorough investigation of this is of course beyond the scope of this paper, I do think it could be worth expanding on this a bit more in the discussion. Clearly more work (both empirical and with simulations) will need to be done in the future to investigate this topic, and one nice contribution of this paper could be to stimulate further discussion of this issue.

Response to Reviewer

Comments: Thanks to the authors for their responses and revisions regarding the point I raised in the previous review. I agree with them that the strongest evidence in favor of metric vs topological interactions in the two contexts comes from the anisotropy analysis, and as stated in my previous review I find this analysis well done and supporting of their claims. I do think they have slightly misinterpreted my previous point regarding the alignment angle analysis, which I tried to illustrate through a toy model simulation. The simulation I made was not meant to capture all of the features of the system, rather it was meant to show that the main piece of evidence the authors use to back their claims about topological and metric interactions with respect to alignment angle, i.e. Figure 1c, is possible to reproduce in a manner that doesn't involve a difference between metric and topological interactions but rather just results from a geometric effect of birds turning back toward the predator in one case and not the other. Of course I do not think that the birds are simply flying around in circles - this is obviously not true. My point is that flying around a focal point is different than flying straight when it comes to alignment angles, and that this turning effect could lead to similar results to those shown in Figure 1c. As a side note, the authors point out that my simulation doesn't capture the velocity distribution, and in particular the fact that the radial speed is of the same order of magnitude as the overall speed. Of course it doesn't, because this was a toy example meant to illustrate my point about Figure 1c, not to capture all aspects of the system. In fact I've set all bird speeds to 1 in the simulation, and depending on the noise added to the velocities, the radial speeds can range from close to 0 to close to 1.

Anyway, I think the authors have taken my main point that the anisotropy analysis is really the thing that can distinguish between metric and topological interactions in the paper, not the alignment angle analysis. I think we still disagree slightly on whether the alignment angle analysis is providing any insight into this issue (I think it really does not, the authors still seem to think it does). I'm fine with them leaving that analysis in, as long as they are clear about its limitations in the text. The sentences: "It is difficult to separate the effects of local interactions and external factors on alignment angles. Thus, to more clearly discriminate between the interaction rules in mobbing and transit flocks, we consider the spatial distribution of neighbours relative to a given focal bird." go in this direction, which I think is a good addition.

Response: We appreciate the reviewer's willingness to compromise on these points.

Comments: More generally, think that the question of how we can rule out effects of external influences on the inferences we draw regarding interaction type is an important one for this field - though a thorough investigation of this is of course beyond the scope of this paper, I do think it could be worth expanding on this a bit more in the discussion. Clearly more work (both empirical and with simulations) will need to be done in the future to investigate this topic, and one nice contribution of this paper could be to stimulate further discussion of this issue.

Response: We fully agree with the reviewer on this point. To highlight it, we added a final sentence to the Discussion section: "Nevertheless, fully understanding and predicting the effect of ecological context on group movement and function will require additional study to separate the distinct effects of local interactions and external factors."